# Stalagmite paleomagnetic record of a quiet mid-to-late Holocene field activity in central South America

Plinio Jaqueto [1✉], Ricardo I. F. Trindade [1], Filipe Terra-Nova[1], Joshua M. Feinberg [2], Valdir F. Novello [3], Nicolás M. Stríkis [4], Peter Schroedl[5], Vitor Azevedo[4,6], Beck E. Strauss [7], Francisco W. Cruz[8], Hai Cheng [9,10] & R. Lawrence Edwards[5]

Speleothems can provide high-quality continuous records of the direction and relative paleointensity of the geomagnetic field, combining high precision dating (with U-Th method) and rapid lock-in of their detrital magnetic particles during calcite precipitation. Paleomagnetic results for a mid-to-late Holocene stalagmite from Dona Benedita Cave in central Brazil encompass ~1900 years (3410 BP to 5310 BP, constrained by 12 U-Th ages) of paleomagnetic record from 58 samples (resolution of ~33 years). This dataset reveals angular variations of less than 0.06° yr$^{-1}$ and a relatively steady paleointensity record (after calibration with geomagnetic field model) contrasting with the fast variations observed in younger speleothems from the same region under influence of the South Atlantic Anomaly. These results point to a quiescent period of the geomagnetic field during the mid-to-late Holocene in the area now comprised by the South Atlantic Anomaly, suggesting an intermittent or an absent behavior at the multi-millennial timescale.

[1] Instituto de Astronomia, Geofísica e Ciências Atmosféricas, Universidade de São Paulo, 05508-090 São Paulo, Brazil. [2] Institute for Rock Magnetism, University of Minnesota, Minneapolis, MN 55455, USA. [3] Department of Geosciences, University of Tübingen, 72076 Tübingen, Germany. [4] Departamento de Geoquímica, Universidade Federal Fluminense, 24020-141 Niterói, Brazil. [5] Department of Earth Sciences, University of Minnesota, Minneapolis, MN 55455, USA. [6] Department of Geology, Trinity College Dublin, Dublin 2, Ireland. [7] NIST, Gaithersburg, MD 20899, USA. [8] Instituto de Geociências, Universidade de São Paulo, 05508-080 São Paulo, Brazil. [9] Institute of Global Environmental Change, Xi'an Jiaotong University, Xi'an 710054, China. [10] Key Laboratory of Karst Dynamics, MLR, Institute of Karst Geology, CAGS, Guilin 541004, China. ✉email: pjaqueto@gmail.com

The magnetic field of the Earth and its variations through time have been investigated with a panoply of instruments, from simple magnetic compasses aboard 16th century ships, to a network of geomagnetic observatories implemented in the 19th century[1] and lately by satellites covering the whole globe[2]. Analysis of these data since the mid-19th century revealed a low-intensity region in the South Atlantic. This South Atlantic Anomaly (SAA) is nowadays the most prominent expression of the non-dipolar field on Earth's surface[3,4]. Centennial-scale reconstructions of the SAA are fundamental for understanding the origin and persistence of this important geomagnetic feature. Nevertheless, the most common materials used to track past geomagnetic field variations are archeological artifacts, volcanic rocks, and sediments. These can be divided into two main classes[5]. The first comprises archeological and volcanic materials which acquire thermal-magnetic remanence upon cooling. In the last decade, new entries of archeomagnetic and volcanic data in GEOMAGIA[6] from 400 CE to 1990 CE for South America better constrain the spatial resolution[7], evolution, and emergence[8] of the SAA (86 new entries from Argentina: 38, Brazil: 32, Chile: 8, Colombia: 5, Ecuador: 3). They provide spot readings of the field's absolute paleointensity and directional data when the original orientation of archeological structures and rocks can be assessed, but they rarely provide continuous stratigraphic records. The second material type comprises sediments whose natural magnetism is acquired during deposition or shortly thereafter (post-depositional magnetic remanence). Sediments provide a continuous record of the variation of the geomagnetic field, are faithful recorders of its directional behavior, and may also provide relative paleointensity estimates if certain conditions are satisfied[9]. In South America, there are only a few sedimentary records from Argentinian lakes[10–14] that have been included in the construction of geomagnetic field models for the Holocene[3,15] and late Quaternary[16]. These records have an average sedimentation rate of 77 cm/ka (from 30 cm/ka[10] to 207 cm/ka[12]) and provide decadal to centennial resolution. Such studies are essential benchmarks for understanding geomagnetic field variations in the Southern hemisphere[15], and in the context of the SAA they capture a more active regime of secular variation in the Southern Hemisphere, especially in South America[3,12]. However, their depositional age may not correspond to the age of remanence acquisition due to the time difference between sedimentation and the lock-in of magnetic particles in sedimentary strata[9,17] and many sediments experience post-depositional alteration of their magnetization due to compaction and chemically-induced dissolution and/or precipitation. Furthermore, the scarcity of continuous high-resolution geomagnetic field reconstruction from South America prevents us from assessing consistency among such records.

A combination of records from archeological artifacts, volcanic rocks, and sediments is used to construct geomagnetic field models. Due to their time coverage, sedimentary records are key to understanding the geomagnetic field's time evolution at the centennial to millennial timescales[18]. For example, these models reveal a persistent and dominant westward drift at high latitudes[19] and a dominant 1350 year cycle in the dipole tilt variation for the past 9000 years[20]. It has also been shown for the last 10,000 years that the southern hemisphere has a weaker average field strength than the northern hemisphere. Furthermore, the Atlantic hemisphere has more active secular variation than the Pacific hemisphere[3]. Nevertheless, data coverage is described as a limiting factor when attempting more refined models, especially to understand short-time variations at high resolution in the southern hemisphere[3,15,18,21].

Speleothems are chemical sediments formed in caves and their global distribution provides an opportunity to improve the geographic coverage of paleomagnetic studies. Speleothems can often be dated precisely using U-Th methods and growth rates allow researchers to obtain paleomagnetic records with resolutions between decadal to millennial timescales[22]. Directional data from speleothems have been used to study the timing and structure of geomagnetic field excursions when the virtual geomagnetic pole locally departs more than 45° from its time-average position[23–26]. Also, speleothems are a promising material to study the recent secular variation of the geomagnetic field at high temporal resolution[27–29]. In particular, a speleothem from central South America covering the last 1500 years demonstrated high rates of angular variation (>0.1°/yr) and intensity drops with a time lag of ~200 years when compared to equivalent events in South Africa[30,31]. These records were interpreted as a result of the recurrence of the South Atlantic Anomaly as it migrates westward (and southward), combined with its expansion and intensification. Here we present a paleomagnetic study of a speleothem from Dona Benedita cave, in central Brazil, with ages within 3410 BP to 5310 BP BP. This study reports U-Th ages, paleomagnetic directions, relative paleointensity, and an assessment of the magnetic mineralogy of one well-dated speleothem, expanding the record of the geomagnetic field in central Brazil to ~5310 BP.

## Results

**Sampling and U-Th dating**. Dona Benedita cave (20.57°S, 56.72°W) is located in central-western Brazil (Fig. 1), along the karst of Serra da Bodoquena that comprises carbonate and terrigenous rocks from the Corumbá Group (Neoproterozoic)[32]. The present-day climate in the study area is humid tropical with an average temperature between 22 °C and 24 °C, with a three-month-long dry season during the austral winter (JJA) and annual rainfall of ~1419 mm[32]. Vegetation is dominated by woodland and savannah forests that correspond to the Brazilian Cerrado Biome.

The stalagmite DBE50 from Dona Benedita cave is part of the collection of the Instituto de Geociências, Universidade de São Paulo. The sample is a fragment of a candle-like stalagmite 417 mm in height (Fig. 1). A total of 58 paleomagnetic specimens were cut using a diamond wire saw to avoid loss of material. The average size of the specimens is ~6.9 mm in height, ~14.7 mm in length, and ~11.1 mm in width. Specimen sizes were chosen to maximize the signal-to-noise ratio and to avoid growth layer curvature along the exterior walls of the speleothem. For the magnetic mineralogy study, powder samples of ~0.03 g were prepared. The stalagmite DBE50 covers ~1900 years, from 3410 BP to 5310 BP, with an average specimen resolution of ~33 years (Fig. 1). The high $^{232}$Th concentration of the speleothem, with an average of ~9963 ppt, is a result of its high detrital content. This value provides a low $^{230}$Th/$^{232}$Th ratio with an average of ~3.4 × 10$^{-5}$; typical U-Th dating assumes an initial $^{230}$Th/$^{232}$Th ratio of 4.4 × 10$^{-6}$ ± 2.2 × 10$^{-6}$[33]. The low values found for $^{230}$Th/$^{232}$Th ratios in this speleothem are expected in a sample with visible clay and silt layers (Fig. 1) and imply a relatively high error in age determinations (2σ error of ~434 years). From the 14 U-Th analyses, only 2 age points were discarded in the geochronology model due to age inversions (detected by StalAge algorithm) (Supplementary Information, Table S1), and the main age model was generated with the remaining age estimations using the StalAge algorithm[34]. Also, to test whether major changes could occur by a different selection of dating points, different scenarios were generated (Supplementary Information, Figures S1) using StalAge[34] and COPRA[35] algorithms, and a major linear trend is observed, so the exclusion of only 2 inversion points was a better choice (Supplementary Data 1).

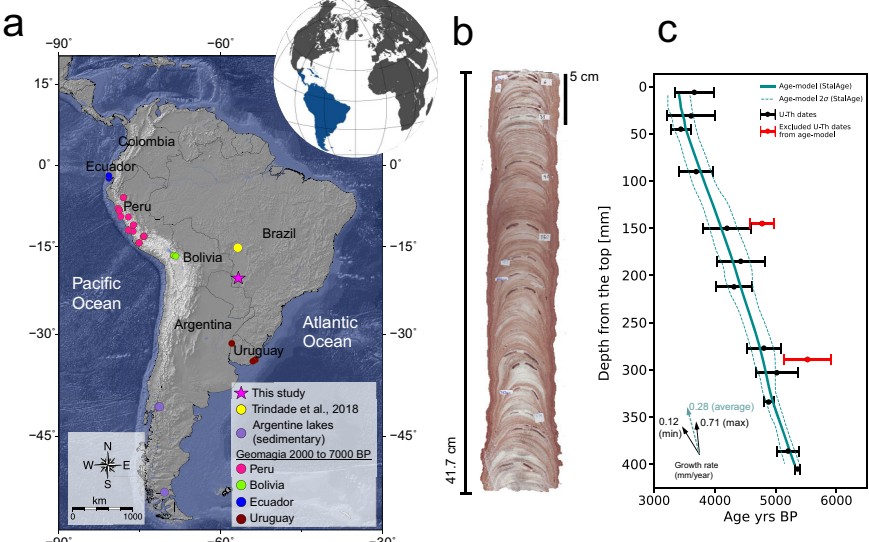

**Fig. 1 Location, sample and age-model of Dona Benedita cave. a** Location of Dona Benedita cave (pink star) and Pau d'Alho cave[29] (yellow circle). Also represented are the records from lakes (purple circles)[10–12], archeointensity and volcanic rocks for ages between 2000 BP and 7000 BP available in the GEOMAGIA50 database v3.4[50] derived from Ecuador (blue)[51,52], Peru (pink)[53–57], Bolivia (green)[51] and Uruguay (brown)[58] (Map generated with GMT software[76]). **b** Stalagmite DBE50 from Dona Benedita cave and **c** age model obtained through U/Th dating; data in red are inverse ages that were discarded in the age model. The solid cyan line represents the age model obtained with the StalAge algorithm[34] and dashed lines represent 95% confidence intervals. Arrows represent growth-rates with minimum and maximum values in solid black and average value in dashed cyan.

**Rock magnetism.** Rock magnetic low-temperature experiments included field-cooled (FC) and zero field-cooled (ZFC) measurements, and room-temperature saturation remanent magnetization (RTSIRM) were performed on magnetic extracts obtained from the samples. In addition, bulk rock samples were imparted with stepwise anhysteretic remanent magnetizations (ARM), and the corresponding acquisition curves were deconvolved to identify different magnetic components.

FC-ZFC experiments show that fine particles dominate the magnetic signal, as magnetizations during field cooling are stronger than those at equivalent temperatures during ZFC (Fig. 2). They also indicate the presence of goethite by the separation between magnetization curves at all temperatures[36]. The Verwey transition (~120 K) is often minimized or entirely suppressed in FC-ZFC curves, and this phenomenon can be attributed to the maghemitization of the original magnetite particles[37].

The presence of magnetite is confirmed in RTSIRM experiments by the Verwey transition as a prominent drop in magnetization at ~120 K during cooling (Fig. 2). The presence of both maghemite and pure stoichiometric magnetite suggests a partially oxidized magnetite core with a maghemitized shell, a common feature of magnetic particles found in soils[38,39].

Finally, the median destructive field (MDF) of ARM for sample DBE50[40] shows values of ~16 mT (low-coercivity) and a dispersion parameter of ~0.28, consistent with extracellular magnetite of pedogenic origin[41], also suggesting the transport of these particles from the soil in the epikarst into the cave through drip water. This pedogenic magnetic fingerprint is common in other speleothems studies[42–44].

**Paleomagnetism.** Paleomagnetic directions were isolated after alternating field (AF) demagnetization between steps 8 mT and 30 mT (Fig. 3, Supplementary Data 2). The magnetic stability of the characteristic component is supported by its internal coherence and by the low values of maximum angular deviation (MAD) and deviation angle (DANG)[45]. The mean MAD found

was ~3.8° and the mean DANG was ~2.8° (Fig. 4c). Compared to the MAD, the lower value of DANG indicates that the characteristic magnetic component points to the origin[46]. Inclinations show good agreement with the expected Geocentric Axial Dipole inclination for the site (−36.9°). The Fisher mean declination and inclination values are −7.9°, and −41.0° ($N = 56$), respectively, with an $\alpha_{95}$ ~1.6°, with a filtered MAD of 8° (Fig. 3, Supplementary Data 2).

Results were plotted as timeseries and compared with different geomagnetic models after rotation of the declinations to the mean declination of model CALS10k.2[3] for the same period (Fig. 4). The geomagnetic models chosen for comparison comprise the period covered by our data: CALS10k.2[3], HFM.OL1.A1[3], pfm9k.1a[47], SHA.DIF.14k[48] and BIGMUDI4k.1[49], the last one being the most updated but limited to the last 4000 years (Fig. 4).

Paleomagnetic directions show good agreement with the models. Magnetic inclination (Fig. 4a) shows a decrease from −35° to −50° at the beginning of the record (5500 BP to 5000 BP), following the trend of models CALS10k.2 and HFM.OL1.A1, although inclination from the stalagmite is about 10° lower than these models between 5500 to 4500 BP. The model pfm9k.1a presents higher values for inclination than the speleothem but with a similar trend, except for a peak at ~4800 BP. The model SHA.DIF.14k differs significantly from the data and other models with peaks of low inclination at ~3200 and ~4800 BP. After ~5000 BP, the speleothem record shows a linear increase of inclination from −50° to −30° until 3000 BP (Fig. 4), which agrees with the tendency of all models and remarkably matches the BIGMUDI4k.1 (from 4000 BP onwards). Declination results (Fig. 4b) shows a westward trend at the beginning of the stalagmite record, between 5500 BP to 4200 BP, which is not in perfect agreement with the geomagnetic models that tend to be flat at the same time interval. Then, a short-term (800 year) eastward trend is observed from 4500 BP to 3800 BP, followed by a westward trend from 3800 BP to 3000 BP. This pattern agrees with the amplitudes observed in the BIGMUDI4k.1 model and follows the HFM.OL1.A1 as well, but it is not predicted by the CALS10k.2 nor the SHA.DIF.14k models.

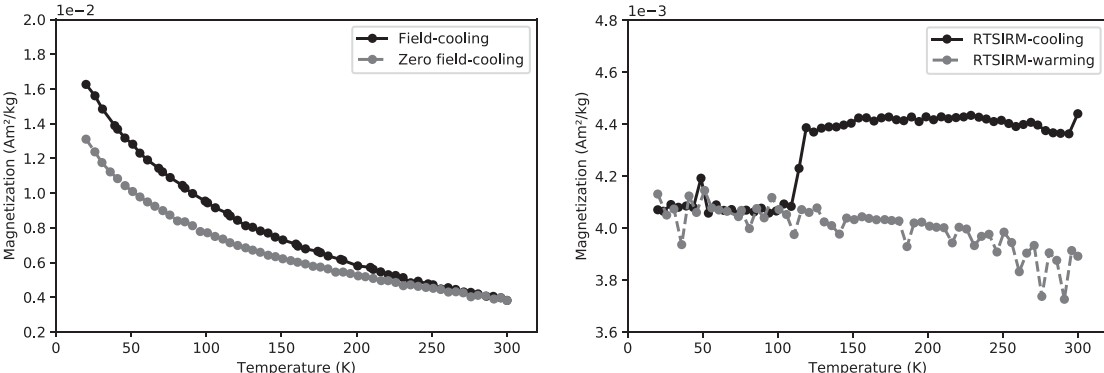

**Fig. 2 Low-temperature curves after magnetic extracts.** Magnetic mineralogy from magnetic extracts of stalagmite DBE50; *(left)* Curves of Field-cooling (2.5 T) (black circles) and Zero-field cooling (gray circles) measured on warming; *(right)* Curve of room temperature saturation isothermal remanent magnetization (RTSIRM); the Verwey transition (~120 K) which is characteristic of magnetite is signaled by the drop observed in the cooling curve (black circles).

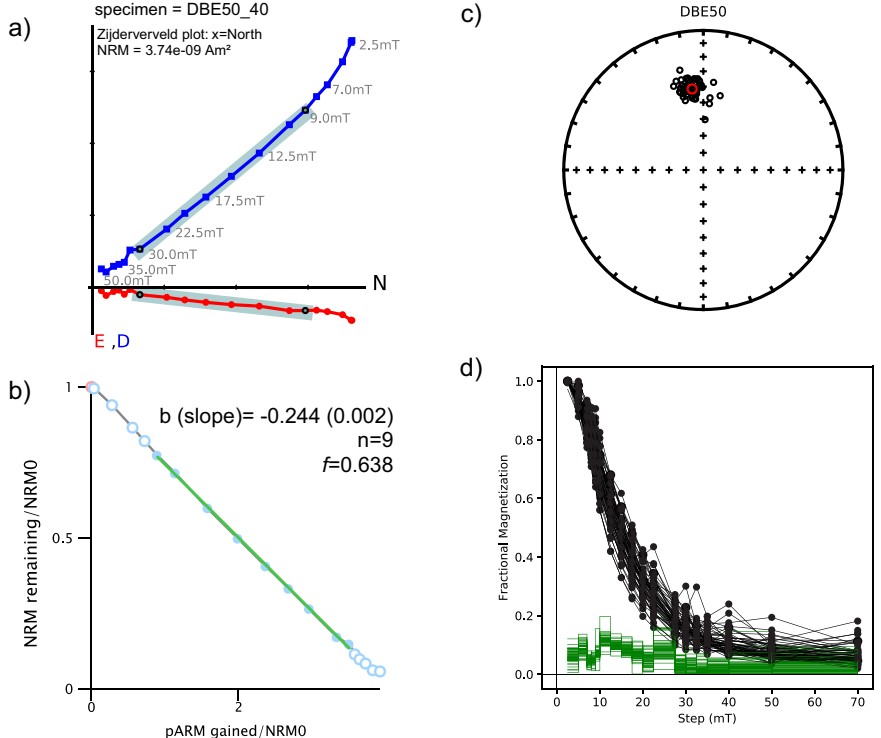

**Fig. 3 Summary of paleomagnetic directions and paleointensity obtained.** Paleomagnetic results of DBE50 stalagmite; **a** orthogonal vector plot for specimen DBE50_40, with respective horizontal (red) and vertical (blue) components and the characteristic direction in cyan shade (from 9 mT to 30 mT). Directional data was obtained using the PMAGPY software[73]. **b** Relative paleointensity for the same specimen calculated with the pseudo-Thellier technique along the same steps ($n = 9$), corresponding to a magnetization fraction of 63% ($f = 0.63$). The slope of the Arai-plot was obtained in Paleointensity.org software[75]. **c** Equal area plot of characteristic directions for all specimens of the DBE50 stalagmite. **d** Demagnetization plot obtained after stepwise alternating field up to 70 mT for all DBE50 specimens (black curves), and their respective gradient (green lines) showing the preponderant contribution of magnetic fraction with remnant coercivity between 10 mT and 30 mT.

Relative paleointensity (RPI) estimates were obtained using the pseudo-Thellier method applied to an average fraction of $0.58 \pm 0.07$ of the natural remanence, typically encompassing eight demagnetization steps. The mean best-fit slope to the resulting Arai diagrams was $-0.19 \pm 0.01$ (Fig. 3, Supplementary Data 2). These results were then normalized by multiplying the absolute value of the slopes by the median value of 191.1 (See Methods for further information). A cubic spline with a 75 year knot was calculated to plot the RPI curve for the DBE50 stalagmite (Fig. 5). The RPI results display a high variability compared to the geomagnetic field models. However,

they are compatible with the range of absolute GEOMA-GIA50.v3.4 datapoints[50] for South America derived from Bolivia[51], Ecuador[51,52], Peru[53–57], and Uruguay[58] (Fig. 5). A decrease in intensity at the beginning of the stalagmite record from 5500 BP to 4500 BP is observed, followed by a higher intensity peak between 4500 BP and 4000 BP, and finally, a higher variability with a median 10 μT drop towards lower intensities is observed from 4000 BP to 3000 BP. The average trend defined by the speleothem record agrees with the models between 5500 BP and 3500 BP. Nevertheless, it is significantly lower for the younger record segment between 3500 BP and 3000 BP.

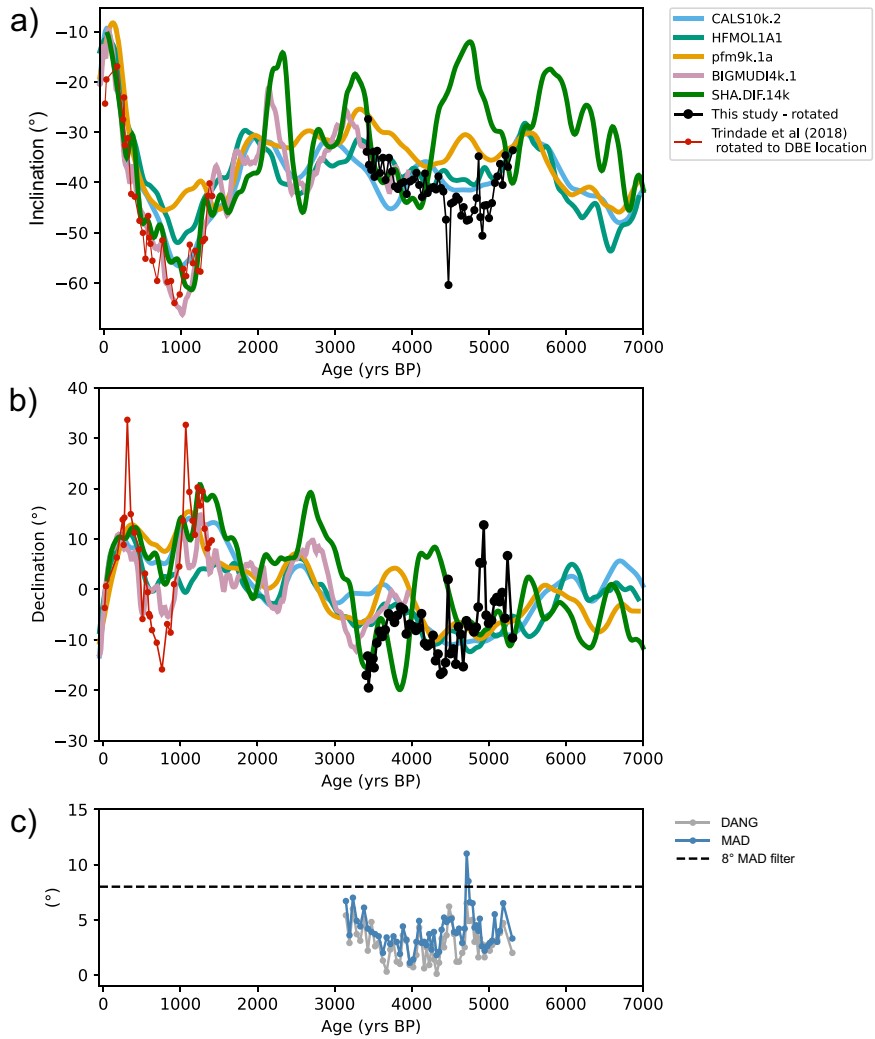

**Fig. 4 Summary of paleomagnetic directions obtained.** Magnetic direction data for stalagmite DBE50 (black dots): **a** inclination, **b** declination, **c** maximum angular deviation (MAD), and Deviation angle (DANG). Also represented are the following geomagnetic models for the location of Dona Benedita cave: CALS10k.2 (light blue curve), HFM.OL1.A1 (green curve), pfm9k.1a (gold curve), BIGMUDI4k.1 (light pink curve), SHA.DIF.14k (dark green curve). Results obtained for stalagmite ALHO6 (Pau D'Alho cave, red dots) were relocated to the DBE50 site location. An 8° filter (dashed line) was applied for MAD values of DBE50 specimens.

## Discussion

The DBE50 speleothem contains tiny amounts of partially oxidized magnetite likely originating from pedogenic processes in the soil above the cave, as well as some amount of goethite, as found in other speleothem examples worldwide[22,29,40,42–44]. In contrast to lake sediments, the magnetization acquisition in speleothems is faster, and they seem to be devoid of any post-deposition effects[22]. As a result, the relationship between radio-isotopic dating and the age of magnetization acquisition is also more straightforward. The homogeneity of magnetic mineralogy observed in this speleothem and other speleothems favor more robust relative paleointensity estimations. The high-quality directional and paleointensity record of DBE50 with average growth rate of 0.28 mm/yr expands on the previous records of Pau d'Alho cave speleothems[29] collected in the same region, and which have similar growth rates of 0.17 mm/yr[29]. These younger speleothems grew over the last 1500 years and reveal a progressive drop in field intensity and geomagnetic field variations linked to the SAA (Fig. 5) with angular velocities higher than 0.1° yr$^{-1}$ in two different time intervals at ~1050 BP and ~500 BP (Fig. 5). In contrast, the DBE50 stalagmite data reported here shows more limited geomagnetic field variability: the median intensity

difference is ~10 μT (±13 μT) and the angular velocities are lower than 0.06° yr$^{-1}$ for the interval between 3150 BP and 5310 BP. This low angular variability is predicted by almost all models, except for the SHA.DIF.14k (Fig. 5). Taken together, the DBE50 results reveal a period of low secular variation activity during the mid-to-late Holocene in central South America (Fig. 5). Our records show that the geomagnetic behaviors associated with the occurrence of the South Atlantic Anomaly, namely low-intensities and directional variations >0.1° yr$^{-1}$, have no counterpart during the mid-to-late Holocene.

The South Atlantic Anomaly is usually attributed to the motion and intensification of geomagnetic reverse flux patches (RFPs) at the core-mantle boundary (CMB)[31,59–61]. These features arise from the expulsion of toroidal field lines by diffusion due to flow upwelling at the top of the core[62]. Normal flux patches (NFPs), on the other hand, arise from the concentration of poloidal field lines by downwelling at the edge of the tangent cylinder[63]. Furthermore, geomagnetic flux patches are responsible for a significant North-South hemispherical asymmetry in the advective sources of the axial dipole moment observed nowadays[59,64]. Ideally, in a purely axial dipole field, the minimum intensity is located along the geographic equator. However, the contribution

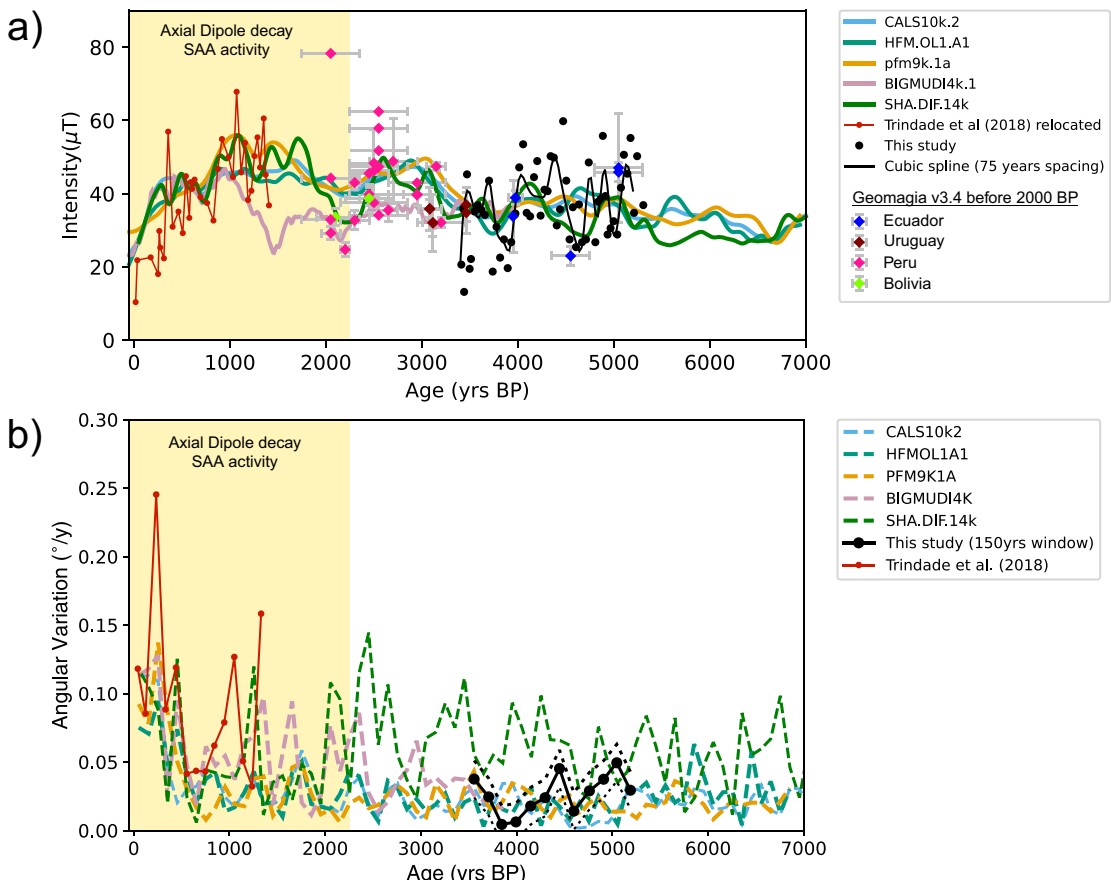

**Fig. 5 Paleointensity and angular variation comparison of obtained data and geomagnetic field models.** Geomagnetic field variations in central Brazil.
**a** Relative paleointensity data for stalagmite DBE50 (black dots) and cubic spline fit for 75 years knots (black curve). Models: CALS10k.2 (light blue curve), HFM.OL1.A1 (green curve), pfm9k.1a (gold curve), BIGMUDI4k.1 (light pink curve), SHA.DIF.14k (dark green curve). Results from ALHO6 (Pau D'Alho cave)[29]; relocated to the Dona Benedita cave coordinates are shown as red dots. Absolute intensity and respective errors (bars) from South America were obtained from Geomagia v3.4[50] derived from Bolivia (green)[51], Ecuador (blue)[51,52], Peru (pink)[53–57] and Uruguay (brown)[58]. **b** Angular variation within a 150 yr window for directions from the Dona Benedita stalagmite (black dots) and Pau d'Alho cave stalagmite (red dots) showing the contrasting behavior before and after the geomagnetic dipole decay[18] and the likely onset of SAA in South America[29] (yellow shade).

of non-axial dipole field components may lead to a departure from this ideal case resulting in complex field morphologies, i.e., a weaker dipole field may result in a more prominent contribution of localized non-dipolar features such as RFPs and NFPs, leading to a significant dislocation of the field minima away from the geographic equator.

To assess the evolution of RFPs and NFPs through the last 10,000 years, we identified these features in models CALS10k.2, HFM.OL1.A1 and pfm9k.1a. Similar analyses were conducted for the last 3,000 years for NFPs[65] and for RFPs[66,67]. Figure 6 shows our results for model CALS10k.2 (results for other models in Supplementary Information, Figures S2, S3). A marked contrast is observed in the occurrence of RFPs for three different time intervals (Fig. 6a–c). Interval #1 (50-2150 BP), comprises the time of the South Atlantic Anomaly and other similar recurrent features, and shows several RFPs in the southern hemisphere. The simultaneous tracking of the SAA minimum and RFPs and NFPs through time showed that the position, motion, and amplitude of the anomaly are highly influenced by the interplay between three persistent geomagnetic flux patches: an RFP in the tropical-subtropical South Atlantic, the South Pacific high-latitude NFP and a low-latitude intense NFP near Africa[60]. These flux patches are tracked in the considered models (Fig. 6a). Interval #2 (3000-5100 BP) corresponds to the time growth interval of speleothem DBE50 and shows fewer RFPs. The RFPs that do occur during

Interval#2 are located in the northern hemisphere (Fig. 6b). Finally, interval #3 (6500-8600 BP), shows more frequent RFPs, mainly located at higher latitudes in the southern hemisphere and mid-to-high latitudes in the northern hemisphere (Fig. 6c). Results from models HFM.OL1.A1 (Figure S2) and pfm9k.1a (Figure S3) also show these same contrasting concentrations of RFPs between the time intervals considered. In all models, intense NFPs observed throughout the past 10,000 years follow the azimuthal positions coincident with the positive peak of shear wave velocity anomalies at the lowermost mantle (Fig. 6d) in the seismic model[68], reinforcing the strong mantle control on these features as previously suggested[69]. In latitude, the normal flux-patches are limited by the expected coordinates for the tangent cylinder at 72° (Fig. 6e).

We can now consider the role of the dipole strength and the non-dipole components to the variability of the field for the considered time intervals. Fig 7a shows the spectral power of the dipole and non-dipole components from models CALS10k.2, HFM.OL1.A1 and pfm9k.1a. Interval #1 shows a continuous decrease of the dipole concomitant to a general increase of the non-dipole components, expressed by a relatively high non-dipole/dipole ratio of ~0.02 (Fig. 7b), which is likely the source of the high angular variations observed in the past two millennia. In contrast, interval #2 shows progressively increasing dipole power, the lowest non-dipole/dipole ratios of the time intervals

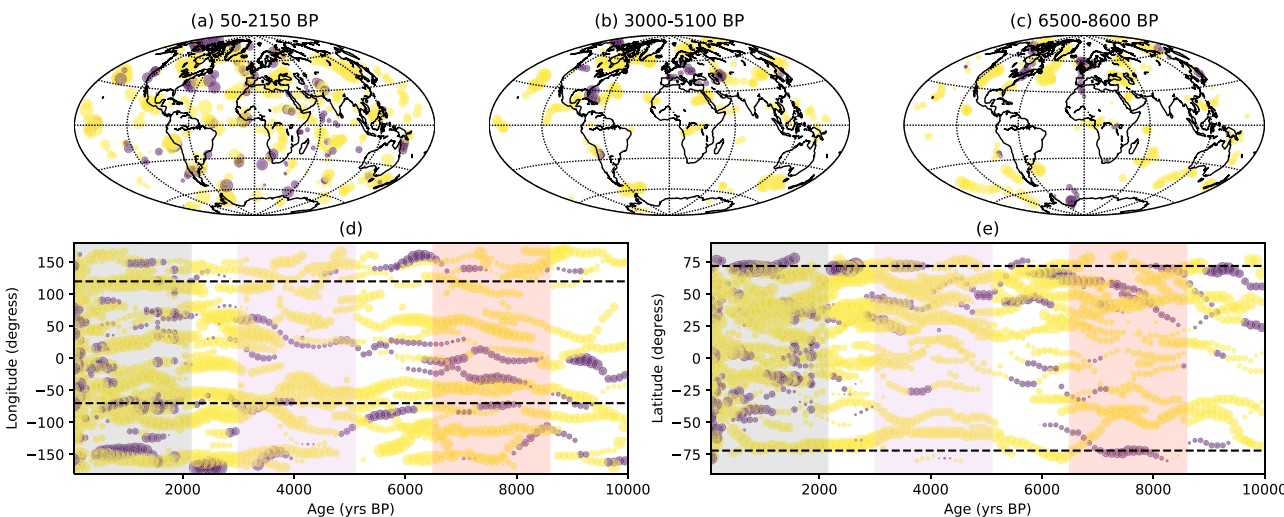

**Fig. 6 Comparison of normal and reversed flux patches of three specific intervals of the Holocene.** Tracking of normal (NFP) and reverse (RFP) flux patches at the core-mantle boundary for model CALS10k.2[3]. The location of NFP (yellow) and RFP (purple) for time intervals #1 (**a**), 50-2150 BP), #2 (**b**), 3000-5100 BP), #3 (**c**), 6500–8600 BP). **d** Longitudinal evolution of NFP (yellow circles) and RFP (purple circles); dashed lines indicate the azimuth of the positive peak of shear wave velocity in the mantle[68], rectangle shades correspond to interval #1 (light grey), #2 (light purple), #3 (light red) **e** latitudinal evolution of NFP (yellow circles) and RFP (purple circles); dashed lines indicate the expected latitude of the tangent cylinder in the Northern and Southern hemispheres. The size of purple and yellow circles is proportional to the intensity of the flux patch. The Dona Benedita record was acquired during the second interval. Map generated using python package Cartopy[77].

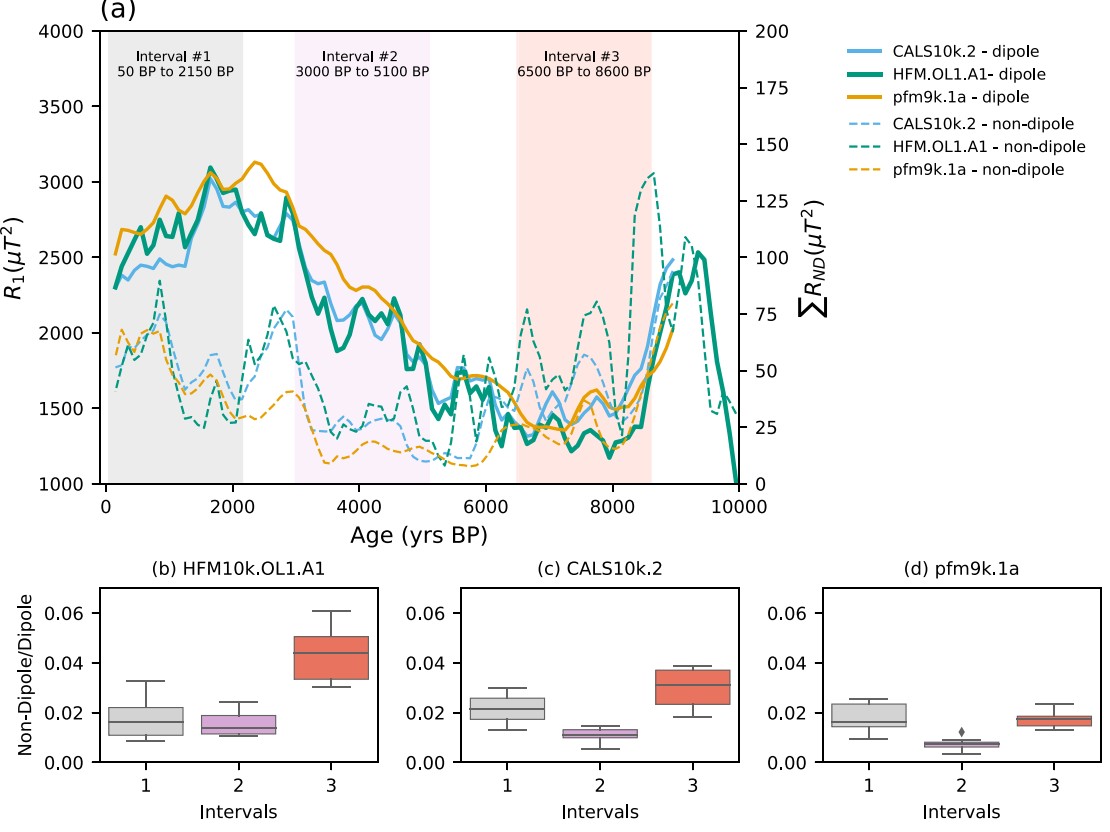

**Fig. 7 Dipolar and non-dipolar energy of the Earth's magnetic field at surface during Holocene. a** Energy at Earth's surface of dipole and non-dipole moments for the past 10,000 years for models CALS10k.2 (light blue curve), HFM.OL1.A1 (green curve), pfm9k.1a (gold curve), with selected time intervals #1 (50-2150 BP, light grey), #2 (3000-5100 BP, light purple) and #3 (6500-8600 BP, light red). box-plots for non-dipole to dipole ratios for time intervals #1, #2 and #3 for models (**b**) HFM.OL1.A1[3], (**c**) CALS10k.2[3] and (**d**) pfm9k.1a[47]. The energy of the dipole and non-dipole and its subsequent ratio was calculated from the Gauss coefficients of the models up to 10 degrees[78].

considered (Fig. 7b–d), and corresponds to the lowest angular variations recorded in the Dona Benedita speleothem (Figur In interval #3, all analyzed models show a non-dipole/dipole ratio increase to values higher than 0.02 (Fig. 7b–d). The DBE50 stalagmite, therefore, grew during a time of quiet geomagnetic field activity in the South Atlantic and South America, which coincided with an interval of limited RFPs in the southern hemisphere, when the non-dipole field components were less prominent relative to the total field. Generally, our results suggest that the occurrence of South Atlantic-like features along the mid-latitude belt of the South Atlantic is an intermittent phenomenon whose expression at the surface depends on the ratio between the dipole to non-dipole field components, as it reflects the existence of reversed flux in the southern hemisphere.

## Methods

**U-Th dating and age model**. Radioisotopic dating by the U-Th method was done at the Isotope Laboratory of the University of Minnesota (USA) and Xi'an Jiaotong University (China). A total of 14 U-Th ages were obtained from powder samples (~100 mg) following stratigraphic horizons of the speleothem. The chemical procedure for separation of uranium (U) and thorium (Th) follows the procedure described in Edwards et al.[70], and the analysis was performed in a multi-collector inductively coupled plasma mass spectrometer [NEPTUNE (Thermo-Finnigan)], following the methodology available[33]. The final age model for DBE50 stalagmite was calculated using the algorithm StalAge[34].

**Low-temperature remanence experiments**. Stalagmites usually have a low concentration of magnetic minerals, so the preferred method for low-temperature experiments is to first separate the magnetic mineral assemblage from the carbonate matrix[36,71]. This is accomplished by dissolving the carbonate in a mildly acidic buffer solution (pH ~4), followed by a flask extraction method using a Nd magnet and an orbital shaker to extract the magnetic minerals[71].

Extracted magnetic minerals were examined using two protocols to measure low-temperature magnetic properties in a Quantum Design Magnetic Properties Measurement System (MPMS-XL) instrument with a sensitivity of ~$10^{-11}$ Am$^2$ at the Institute for Rock Magnetism (IRM) at the University of Minnesota. The first protocol consisted of applying a 2.5 T field during cooling from room temperature to 10 K (Field Cooled (FC)). The magnetic moment is then measured in 5 K steps during warming up to room temperature in a zero-field environment. After this cycle, the specimen is cooled down to 10 K in a null field, and a 2.5 T field is imparted at 10 K (Zero Field Cooled (ZFC)). The magnetic moment is then measured in 5 K steps during warming up to room temperature. This protocol has been used to identify the presence of goethite (separation between FC-ZFC curves) and low-temperature magnetic transitions, like the Verwey transition ~120 K for magnetite and Morin transition ~260 K for hematite, and also as a grain-size indicator for magnetite and its oxidation state[36,37,44]. The second protocol is the room-temperature saturation isothermal remanent magnetization (RTSIRM), where a pulsed field of 2.5 T is applied at room temperature, and remanence is measured during cooling (300 K to 10 K) and warming (10 K to 300 K) at 5 K steps. The RTSIRM protocol examines only those magnetic minerals that hold remanence at room temperature. It is sensitive to stoichiometric magnetite (Verwey transition) and its oxidation state[36,37]. Also, goethite has been recognized by the increase in magnetization as the temperature cools by a factor of two in RTSIRM experiments[52].

**Paleomagnetism**. Rock magnetic experiments were carried at the Institute for Rock Magnetism (IRM) at the University of Minnesota in a magnetically shielded room with a noise field of less than 300 nT. Remanence measurements were made using a u-channel superconducting magnetometer (2G Enterprises) with an inline alternating field demagnetization device, with a noise field of less than ~$3.0 \times 10^{-11}$ Am$^2$. Demagnetization and acquisition of anhysteretic remanent magnetization (ARM) were conducted over 25 steps up to 70 mT. For the ARM acquisition, a steady field of 0.05 mT was applied along with an alternating field following the pseudo-Thellier protocol[72].

The analysis of directions was made with the PmagPy software[73] to obtain the characteristic remanent magnetization direction (ChRM) using a routine for principal component analysis (PCA)[74]. For relative paleointensity estimates, the slope obtained with ARM$_{gained}$ by NRM$_{left}$ (Arai plot) was calculated using the line fitting method with the software Paleointensity.org[75]. Because the sample was not azimuthally oriented in the field, the procedure adopted was to calculate the Fisher mean for the directional results and compare with the geomagnetic field model CALS10k.1b for the same period, then the difference in mean declination between them was used to rotate the declination results.

To compare the relative paleointensity (RPI) results with the absolute paleointensity record, the data is normalized following the calibration method used in the CALS7k.2 model[17], by multiplying the median ratio of the geomagnetic field model CALS10k.1b by the slope of the pseudo-Thellier method. The angular variation is calculated using a running mean with a 150 years window (encompassing an average of three specimens) for the angular distance between the directions divided by the time interval[29,30].

**Identification of magnetic flux-patches at the core-mantle boundary**. We identified the center of flux patches, both normal (NFP) and reversed (RFP), in geomagnetic field models CALS10k.2, HFM.OL1.A1[3] and pfm9k.1a[47] by defining the local maxima and minima of the radial magnetic field at the core-mantle boundary. Following a methodology previously used in archeomagnetic field models[66], we assign a patch as normal or reversed based on its polarity to the axial dipole and its relative position to the magnetic equator. However, here we use a different approach for the identification of the magnetic equator. We identify all null-curves of the radial magnetic field at the core-mantle boundary and assign the magnetic equator to the one present at least once in all longitudes. This updated method is more robust than the previous strategy and fails only if the magnetic equator reaches the geographic poles. Also, no filtering technique was applied as it is different to previous geomagnetic patches identifications[65,66].

## Data availability

The paleomagnetic data generated in this study have been deposited in the MAGIC database (https://earthref.org/MagIC/19484) and in the Supplementary Information.

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

## Acknowledgements

We are thankfull to Centro Nacional de Pesquisa e Conservação de Cavernas (CECAV/ICMBio) for providing permission to collect stalagmite samples. We are grateful to Augusto Auler and Bruna M. Cordeiro for guiding the first field trip to collect speleothems in Dona Benedita Cave. This study was financed in part by the Coordenação de Aperfeiçoamento de Pessoal de Nível Superior – Brasil (CAPES) – Finance Code 001. This work was supported by the São Paulo Research Foundation (Grants #2016/24870-2, #2016/15807-5, #2017/50085-3, #2018/15774-5, #2018/07410–3 and #2019/06709-8) the Serrapilheira Institute (grant number: Serra-1812-27990) and the National Council for Scientific and Technological Development (CNPq) (Grant numbers: 308769/2018-0 to N.M.S and 426258/2016-9 to ITP). This work was supported by National Natural Science Foundation of China, NSFC 41888101 to HC. This work was also supported by the National Science Foundation grant EAR-2044535 and US-Israel Binational Science Foundation grant #2016402 to JMF. The IRM is a US National Multi-user Facility supported through the Instrumentation and Facilities Program of the National Science Foundation, Earth Sciences Division, and by funding from the University of Minnesota. This work has been partially performed at USPMag laboratory at Instituto de Astronomia, Geofísica e Ciências Atmosféricas (IAG) from Universidade de São Paulo (USP) funded by CAPES/FAPESP/CNPQ. Certain commercial equipment, instruments, materials, and software are identified in this paper to foster understanding. Such identification does not imply recommendation or endorsement by the National Institute of Standards and Technology, nor does it imply that the materials or equipment identified are necessarily the best available for the purpose.

## Author contributions

P.J. designed the study, performed the analyses, and write the original draft. R.I.F.T. and J.M.F. designed the study. F.T.N. performed flux-patches analyses. V.F.N., P.J., and F.W.C. performed field trip. P.J., V.F.N., P.S., V.A. performed dating analyses supervisioned by H.C. and R.L.E. J.M.F. and B.S. performed low-temperature magnetic analyses. F.W.C. and N.M.S. provided funding. P.J., V.F.N., N.M.S,. and V.A. constructed the age model. All the above authors contributed to the final version of the manuscript.

## Competing interests

The authors declare no competing interests.
