## [Peer Review File · Nature Communications]

Stalagmite paleomagnetic record of a quiet mid-to-late
Holocene field activity in central South AmericaReviewers' Comments:

Reviewer #1:

Remarks to the Author:

I welcomed the opportunity to review this paper on the Dona Benedita cave sediments which indicate a quiet geomagnetic field activity during the mid Holocene in Central America.

The results are well presented, scientifically sound and in detail. The directional and relative intensity data are discussed in the context of better defining the South Atlantic Anomaly and similar features. The study robustly discuss and compare the Dona Benedita speleothem data with the existing literature in terms of global geomagnetic models and field structures of Normal and Reversed Patches in the core mantle boundary.

The conclusions are supported by a robust set of data and arguments indicating that the South Atlantic Anomaly is probably a feature with an intermittent behavior at the multi-millennial timescale. The work presented meet the expected standards with enough detail provided in the method section. Few minor comments can be found in the annotated PDF.

Reviewer #2:

Remarks to the Author:

Key results

The manuscript provides high-resolution data (40 yrs per sample) for a quiescent period of regional geomagnetic field over a period of about 2000 years in Brazil (5310 - 3150 years ago). The location is interesting because it is located near the South Atlantic Anomaly, which is today's most outstanding feature of the modern geomagnetic field. The time period is of interest because it was undocumented before and precedes another similar high-resolution record from Brazil (last 1500 years), therefore extending knowledge to ancient times. The authors discuss their results and geomagnetic field models outputs with regards to the physics of the core and conclude that features like the SAA are intermittent phenomenon.

Significance

This kind of high-resolution record from the Southern Hemisphere are extremely useful for the paleomagnetism community and they do not come often. This work is valid and significant, and state-of-the-art methods were used. This work has implications for understanding the geodynamo and also constitutes a dating tool for paleosciences and archaeology in the region.

Data and methodology

Please be consistent with age estimates reports throughout the manuscript. In the intro it is written "...with ages within 3000-6000 BP." and "...expanding the record of the geomagnetic field in central Brazil to ~6000 BP", but in the abstract the older age is 5310 BP. Also, it is unclear to me why the interval 3000-5100 BP was chosen for the discussion (the box in the key figures 6 and 7).

The age estimates could be rounded to the nearest decade. Yearly age estimates are not meaningful within uncertainties. Decade numbers are also easier to grasp for readers. For example in the abstract use 3150 instead of 3147.

Line 226. "new" U-Th ages. Is "new" necessary; are there previously published ages? If there are, add the reference and why were they not considered in the chronology?

Analytical approaches

Lines 81-83. Six U-Th age estimates were discarded because of age inversion. But four of those have age uncertainties that actually overlap (no inversion) with the precedent age. Is an age inversion of the median/mean age sufficient to discard these U-Th data? The total of utilised ages is low (8 out of 14). This could be 12 utilised ages out of 14 for a better use of the data. Or am I missing another good reason to discard those ages? Please expand on the reasons to discard U-Th age estimates in the chronology.

Suggested improvements

Line 216. "Generally, our results suggest that the occurrence of South Atlantic like features along the mid-latitude belt of the South Atlantic is an intermittent phenomenon whose expression at the surface depends on the ratio between the dipole to non-dipole field components, as it determines the existence of the reversed flux in the southern hemisphere." Replace "determines" by "reflects" or "tracks". The ratio does not determine the existence of RFP. The ratio is a proxy that reflects/tracks the existence of RFP. As stated earlier "These features [RFP] arise from [are determined by] the expulsion of toroidal field lines by diffusion due to flow upwelling at the top of the core"

Figure 5. Add a reference for the yellow shade "likely onset of SAA". Add justification for that choice in the main text (likely onset based on what?).

Line 159. "speleothems collected in the same region" maybe add .. "and with similar accretion rates". Maybe add the values in mm/yr for the two stalagmites. These are quite outstanding and worth mentioning.

Line 163. "no significant drops in intensity". In figure 5a I see two intensity drops to about 20 uT between 4 and 3 ka, that are similar in value to the recent intensity drop associated to the SAA (Fig.5a)

Line 199. Remove "are"

Clarity and context

Improve context in consideration of previous work in the introduction

- Line 27. How does the type "archaeomag+volcanic" contribute so far to the understanding of centennial-scale reconstruction of SAA? Provide quantitative info. Example, 135 data in Brazil 425-1990 AD archaeomagnetic data in GEOMAGIA; the hypothetical Holocene excursions of Nami (I'm unsure if they are from archaeomag+volcanic or sediment?).
- Line 35. How does the type "sediment" contribute so far to the centennial-scale reconstruction of SAA? Provide quantitative info on Southern America archives, sedimentation rates and period covered. Examples, Laguna Potrok Aike and other Argentinian lakes. Some of these lakes have >100 cm/ka high sedimentation rates, which corresponds to only a few decades per data point, similar to the 40 yrs per sample of the studied stalagmites.

Figures 1a (legend and figure caption) and 5a (legend) should be "Geomag v3.4 before 2000 BP" (not after). The data shown is from 5.5-2 ka. Some of the GEOMAGIA sites shown on fig1a are >2000 km from the stalagmite site. Maybe color code those sites within 2000 km shown in figure 5 for clarity. Also, I am wondering what is/are the reason/s for choosing 2000 km; adding a sentence explaining

this would be good. Authors can choose different cutoff distance for defining a region, sometimes based on the Earth surface area where the field is assumed to be the same (eg pmag dating purpose) or based on data availability, or else.

References

Line 76. "Reference not found" Please add the reference.

Point-by-point answers

We thank both reviewers for their careful reading of our manuscript, and their detailed comments and suggestions. In this new version, we incorporated all corrections proposed and suggestions for improvement of the work.

The main change in content in the paper concerns the age model:

- 1) New age-model: In order to clarify the reviewers and future readers, we run different scenarios by excluding points that show inverse median age points, this has been added as a Supplementary Figure S1, and cited in the manuscript (more information below). The chosen model was constructed using 12 out of the 14 dating points. Although they present a similar behavior to the original age-model, we think the new model that excludes only 2 points as recommended by one of the reviewers is simpler and more robust. This new age model implied in new figures 01, 02, 04, 05, and in the addition of the Supplementary Figure S1.

“Reviewer #1 (Remarks to the Author):

I welcomed the opportunity to review this paper on the Dona Benedita cave sediments which indicate a quiet geomagnetic field activity during the mid Holocene in Central America.

The results are well presented, scientifically sound and in detail. The directional and relative intensity data are discussed in the context of better defining the South Atlantic Anomaly and similar features.

The study robustly discuss and compare the Dona Benedita speleothem data with the existing literature in terms of global geomagnetic models and field structures of Normal and Reversed Patches in the core mantle boundary.

The conclusions are supported by a robust set of data and arguments indicating that the South Atlantic Anomaly is probably a feature with an intermittent behavior at the multi-millennial timescale.

The work presented meet the expected standards with enough detail provided in the method section. Few minor comments can be found in the annotated PDF.”

We thank the reviewer #1 for the comments and suggestions, we followed all the suggestions made in the annotated PDF, and the changes were incorporated in this new version.

“Reviewer #2 (Remarks to the Author):

Key results

The manuscript provides high-resolution data (40 yrs per sample) for a quiescent period of regional geomagnetic field over a period of about 2000 years in Brazil (5310 - 3150 years ago). The location is interesting because it is located near the South Atlantic Anomaly, which is today's most outstanding feature of the modern geomagnetic field. The time period is of interest because it was undocumented before and precedes another similar high-resolution record from Brazil (last 1500 years), therefore extending knowledge to ancient times. The authors discuss their results and geomagnetic field models outputs with regards to the physics of the core and conclude that features like the SAA are intermittent phenomenon.

Significance

This kind of high-resolution record from the Southern Hemisphere are extremely useful for the paleomagnetism community and they do not come often. This work is valid and significant, and state-of-the-art methods were used. This work has implications for understanding the geodynamo and also constitutes a dating tool for paleosciences and archaeology in the region.

Data and methodology

Please be consistent with age estimates reports throughout the manuscript. In the intro it is written “...with ages within 3000-6000 BP.” and “...expanding the record of the geomagnetic field in central Brazil to ~6000 BP”, but in the abstract the older age is 5310 BP. Also, it is unclear to me why the interval 3000-5100 BP was chosen for the discussion (the box in the key figures 6 and 7).

We thank the reviewer for calling attention to this point. Through the construction of the new age-model, we set the ages for the speleothem between 3410 BP and 5310 BP (an interval of 1900 years). Also, the intervals were changed in the main text to be consistent throughout the whole manuscript. The chosen interval in figures 6 and 7 were done to match the period of investigation of this study (~1900 years), so when scaling the box-plots of dipolar/non-dipolar contributions, we can observe the same rate of changes during the Holocene.

The age estimates could be rounded to the nearest decade. Yearly age estimates are not meaningful within uncertainties. Decade numbers are also easier to grasp for readers. For example in the abstract use 3150 instead of 3147.

Thanks for noting, we made the changes throughout the manuscript and it is clearer now

Line 226. “new” U-Th ages. Is “new” necessary; are there previously published ages? If there are, add the reference and why were they not considered in the chronology?

The stalagmite does not have any previously published ages. We corrected this part and eliminated the “new” in the indicated line and also in introduction.

Analytical approaches

Lines 81-83. Six U-Th age estimate were discarded because of age inversion. But four of those have age uncertainties that actually overlap (no inversion) with the precedent age. Is an age inversion of the median/mean age sufficient to discard these U-Th data? The total of utilised ages is low (8 out of 14). This could be 12 utilised ages out of 14 for a better use of the data. Or am I missing another good reason to discard those ages? Please expand on the reasons to discard U-Th age estimates in the chronology.

We thank the reviewer for raising this issue. Initially we opted for discarding the ages based on the inversion of the median values, but we took the reviewer suggestions and run different scenarios using two algorithms designed to construct speleothem age-models: StalAge (Scholz and Hoffmann, 2011) and COPRA (Breitenbach et al., 2012). Although there are no significant changes on the general behavior of the generated age-models, we decided that the model obtained with the StalAge algorithm considering 12 ages (discarding only outliers #5 and #10) offers a better estimate, due to the straight-line fitting of at least 3 points (COPRA does the fitting using 2 points). The generated scenarios were included in the supplementary material, and this information was added in the manuscript. Also, we are making available all age dataset, where future selections could be done by the readers. Below we show the selected age-model and the different age model scenarios.

Figure 1: Age-model from stalagmite DBE50. (left) Final age model, determined with algorithm StalAge by only excluding points #5 and #10 (red). (right) Different scenarios generated with StalAge (stal), COPRA algorithms and linear fit. The symbol (#) indicate the age points that were excluded from the adjustment for each scenarios.

Suggested improvements

Line 216. “Generally, our results suggest that the occurrence of South Atlantic like features along the mid-latitude belt of the South Atlantic is an intermittent phenomenon whose expression at the surface depends on the ratio between the dipole to non-dipole field components, as it determines the existence of the reversed flux in the southern hemisphere.” Replace “determines” by “reflects” or “tracks”. The ratio does not determine the existence of RFP. The ratio is a proxy that reflects/tracks the existence of RFP. As stated earlier “These features [RFP] arise from [are determined by] the expulsion of toroidal field lines by diffusion due to flow upwelling at the top of the core”

We thank the reviewer for raising this point. The reviewer is right and we made the changes in the manuscript.

Figure 5. Add a reference for the yellow shade “likely onset of SAA”. Add justification for that choice in the main text (likely onset based on what?).

We agree with the reviewer, the information was corrected and added in the figure caption.

Line 159. “speleothems collected in the same region” maybe add .. “and with similar accretion rates”. Maybe add the values in mm/yr for the two stalagmites. These are quite outstanding and worth mentioning.

We thank the reviewer for this kind suggestion. This information was added to the manuscript.

Line 163. “no significant drops in intensity”. In figure 5a I see two intensity drops to about 20 uT between 4 and 3 ka, that are similar in value to the recent intensity drop associated to the SAA (Fig.5a)

We thank the reviewer for raising this point. Although there is a strong variability in our data due to the nature of relative paleointensity calculation, we took the reviewer suggestion and calculated the median value for the period from 3 to 4 ka and from 4 to 5.3 ka. The median values calculated was 29 and 39 μ T respectively, showing a 10 μ T drop, with a standard deviation of 13 μ T. This information was added in the manuscript (lines 162-166).

Line 199. Remove “are”

Done.

Clarity and context

Improve context in consideration of previous work in the introduction

- Line 27. How does the type “archaeomag+volcanic” contributes so far to the understanding of centennial-scale reconstruction of SAA? Provide quantitative info. Example, 135 data in Brazil 425-1990 AD archaeomagnetic data in GEOMAGIA; the hypothetical Holocene excursions of Nami (Im unsure if they are from archaeomag+volcanic or sediment?).

We thank the reviewer for raising this issue, we added this information to account for the new entries in GEOMAGIA. (lines 28-31)

- Line 35. How does the type “sediment” contributed so far to the centennial-scale reconstruction of SAA? Provide quantitative info on Southern America archives, sedimentation rates and period covered. Examples, Laguna Potrok Aike and other Argentinian lakes. Some of these lakes have

>100 cm/ka high sedimentation rates, which corresponds to only a few decades per data point, similar to the 40 yrs per sample of the studied stalagmites.

We thank the reviewer for raising this issue, we added this information in the manuscript. (lines 37-41)

Figures 1a (legend and figure caption) and 5a (legend) should be “Geomagia v3.4 before 2000 BP” (not after). The data shown is from 5.5-2 ka. Some of the GEOMAGIA sites shown on fig1a are >2000 km from the stalagmite site. Maybe color code those sites within 2000 km shown in figure 5 for clarity. Also, I am wondering what is/are the reason/s for choosing 2000 km; adding a sentence explaining this would be good. Authors can choose different cutoff distance for defining a region, sometimes based on the Earth surface area where the field is assumed to be the same (eg pmag dating purpose) or based on data availability, or else.

Thanks for noting, we took the reviewer suggestion and color coded the data based on the country. This modification was done in Figures 01 and 05. Also, the 2000 km radius was done to include the data from South America with no specific geomagnetic reason, we decided to remove the 2000 km selection and discuss all the South American data instead.

References

Line 76. “Reference not found” Please add the reference.

Done.

References cited

Breitenbach, S.F.M., Rehfeld, K., Goswami, B., Baldini, J.U.L., Ridley, H.E., Kennett, D.J., Pruffer, K.M., Aquino, V.V., Asmerom, Y., Polyak, V.J., Cheng, H., Kurths, J., Marwan, N., 2012. COntstructing Proxy Records from Age models (COPRA). *Climate of the Past* 8, 1765-1779.

Scholz, D., Hoffmann, D.L., 2011. StalAge - An algorithm designed for construction of speleothem age models. *Quaternary Geochronology* 6, 369-382.

Reviewers' Comments:

Reviewer #2:

Remarks to the Author:

The authors adequately answered the reviewers comments and I consider the manuscript ready for publication.

Point-by-point answers

We thank both reviewers for their careful reading of our manuscript.

Reviewer #2 (Remarks to the Author):

The authors adequately answered the reviewers comments and I consider the manuscript ready for publication.

Many thanks